# Personality-Based Emotion Recognition Using EEG Signals with a CNN-LSTM Network

**DOI:** 10.3390/brainsci13060947

**Published:** 2023-06-14

**Authors:** Mohammad Saleh Khajeh Hosseini, Seyed Mohammad Firoozabadi, Kambiz Badie, Parviz Azadfallah

**Affiliations:** 1Department of Biomedical Engineering, Science and Research Branche, Islamic Azad University, Tehran 14778-93855, Iran; mo.khajehosseini@gmail.com; 2Department of Medical Physics, Faculty of Medicine, Tarbiat Modares University, Tehran 14117-13116, Iran; pourmir@modares.ac.ir; 3Content & E-Services Research Group, IT Research Faculty, ICT Research Institute, Tehran 14399-55471, Iran; k_badie@itrc.ac.ir; 4Department of Psychology, Faculty of Humanities, Tarbiat Modares University, Tehran 14117-13116, Iran

**Keywords:** emotion recognition, personality traits, deep neural network

## Abstract

The accurate detection of emotions has significant implications in healthcare, psychology, and human–computer interaction. Integrating personality information into emotion recognition can enhance its utility in various applications. The present study introduces a novel deep learning approach to emotion recognition, which utilizes electroencephalography (EEG) signals and the Big Five personality traits. The study recruited 60 participants and recorded their EEG data while they viewed unique sequence stimuli designed to effectively capture the dynamic nature of human emotions and personality traits. A pre-trained convolutional neural network (CNN) was used to extract emotion-related features from the raw EEG data. Additionally, a long short-term memory (LSTM) network was used to extract features related to the Big Five personality traits. The network was able to accurately predict personality traits from EEG data. The extracted features were subsequently used in a novel network to predict emotional states within the arousal and valence dimensions. The experimental results showed that the proposed classifier outperformed common classifiers, with a high accuracy of 93.97%. The findings suggest that incorporating personality traits as features in the designed network, for emotion recognition, leads to higher accuracy, highlighting the significance of examining these traits in the analysis of emotions.

## 1. Introduction

Emotions have generated interest in recent years. There are many emotion studies in various fields of science, such as psychology, human–machine interaction [1], and, particularly, neuroscience. Most of these studies are based on emotion stimulation and the processing of physiological responses. Compared with other types of emotional stimuli, images are more interesting because they are more informative, and most studies report high recognition accuracies. In addition, since the brain plays a crucial role in emotional responses, EEG is commonly investigated more than others [2]. Emotions are highly subjective and influenced by various factors, such as stimulation type, memories, experiences, personality, mood, etc. Many psychological studies show that there is a correlation between personality and EEG [3]. The Big Five Personality Traits Questionnaire (IPIP-BFM-50) is one of the most reliable methods for measuring personality. In this model, personality dimensions are divided into two categories: Extroversion and Neuroticism. Extroversion includes extroverts and introverts. Extroverts are social and have more interaction with the outside world in contrast with introverts. Therefore, extroverts require more emotional stimulation than introverts. Neurotics include stable and unstable. Unstable people are easily influenced by external stimulation and tend to have negative feelings, such as depression and anxiety, or positive feelings, such as mania, while stable people are less affected [4]. Based on an individual’s personality, emotional responses may differ based on specific stimuli. In other words, emotions are influenced by personality. Therefore, it is necessary to identify personality in the field of emotion recognition or other perspectives [5]. The way individuals perceive, process, and express emotions is significantly influenced by their personality traits [6]. The Big Five personality traits, including agreeableness, conscientiousness, neuroticism, extroversion, and openness, were proposed by Lew Goldberg in 1990 [7], and they are crucial in determining an individual’s emotional experiences and reactions. Self-reported personality assessments, such as the five-factor personality test [8,9] and the MBTI personality test [10,11], can be used to identify an individual’s personality traits. In real-world settings, considering personality traits can be essential for determining the effectiveness of emotion recognition systems that use Electroencephalogram (EEG) signals [9,12,13]. Recent studies have shown an increasing interest in using physiological signals to predict personality, as they provide researchers with a better understanding of participants’ reactions during experiments [8,9,12,14]. However, analyzing EEG signals for emotion recognition, while accounting for individual differences in personality traits, is a challenging task. Although EEG signals have many significant potential applications in various domains, especially in affective computing fields, they are complex and vary significantly across individuals and emotions, making it difficult to extract meaningful features for emotion recognition [13,15]. To address this problem, researchers have proposed various methods and models to recognize emotions through EEG signals in computational model problems.

The utilization of machine learning methods in EEG emotion recognition has become prevalent due to their enhanced performance in accurately classifying EEG signals into various emotional states [16]. Zheng et al. [17] used deep belief networks (DBNs) to construct an emotion recognition model based on EEG, while Pandey et al. [18] used a multilayer perceptron neural network. Conventional machine learning approaches that focus solely on the analysis of EEG signals may not fully account for the individual differences in personality traits that affect emotion recognition [19]. This is where the utilization of advanced machine learning techniques can be advantageous. Scholars have delved into the relationship between personality models and emotions, as well as the impact of personality on physiological responses. For instance, Martínez et al. conducted a study that explored the role of EEG and personality traits in emotion recognition through the use of machine learning and feature selection techniques [9]. Similarly, a recent study proposed a personalized EEG emotion recognition approach that considers both personality traits and the spatio–temporal features of EEG signals [12]. Another study used EEG and machine learning to recognize personality traits. They achieved accuracies between 73% and 86.5%, by analyzing emotion related EEG recordings, using brain networks, graph theoretical parameters, and SVMs [13].

The aim of this investigation is to apply a combination of machine learning techniques that take into consideration not only the latent characteristics in brain signals but the unique personality traits of individuals. Convolutional neural networks (CNNs) have been extensively used in computer vision tasks, and they have demonstrated remarkable performance in extracting features from high-dimensional data, such as images and signals [20,21]. In the domain of emotion recognition based on EEG signals, CNNs can be employed to automatically extract features from the raw EEG signals that identify the underlying patterns linked to arousal and valence. These features can, then, be utilized to train a classifier to differentiate emotional states [22]. Salama et al. developed a 3D CNN-based method for emotion detection, utilizing the DEAP dataset and multichannel EEG signals, achieving remarkable recognition accuracies of 88.44% and 88.49% for the valence and arousal categories, respectively [21]. Tripathi et al. presented Deep Neural Networks (DNN) and CNN models for emotion detection based on EEG signals and accomplished classification accuracies of 75.58% and 73.28% for valence and arousal, respectively. The researchers discovered that deep learning techniques outperformed conventional methods based on their experimental results [23].

In the context of emotion recognition, long short-term memory (LSTM) networks can be used to capture the temporal dependencies between the audio, visual, and biological signals, and they can also extract more complex features related to personality traits [24]. This can improve the accuracy of the emotion recognition model by accounting for individual differences in personality traits that may affect the EEG signals. This study aims to employ CNN and LSTM networks to extract EEG features for emotion recognition, combining spatial and temporal features to capture patterns and relationships between different EEG channels over time. Research conducted by Zhang et al. demonstrated that CNN and CNN–LSTM models can extract raw data with high accuracy, while DNN models have fast training speeds and LSTM models capture more complex features related to personality traits, but they are less stable and slow to converge [25]. Chakravarthi et al. explored the use of EEG signals for analyzing emotions and Post-Traumatic Stress Disorder (PTSD). The study proposed a new automated CNN-LSTM with a ResNet-152 algorithm that achieved higher accuracy in emotion analysis compared to existing techniques [26]. Bhardwaj et al. conducted a study using a DeepLSTM network to classify personality traits based on EEG signals collected from participants watching videos, achieving superior classification accuracy compared to other machine learning classifiers, with a maximum accuracy of 96.94% [10].

This article presents a new method for emotion recognition based on EEG signals that combines CNNs and LSTM networks. The proposed approach aims to improve the accuracy of the recognition process by capturing both spatial and temporal dependencies in the EEG signals and considering individual differences in personality traits. The CNNs are used to extract features from the raw EEG signals, while the LSTMs are employed to capture temporal dependencies and extract more complex features related to personality traits. The study conducted EEG recordings for volunteer participants and compared the performance of the proposed approach with existing state-of-the-art approaches for EEG-based emotion recognition [27]. Moreover, specifically, we investigate whether a sequence of images and videos can help the LSTM network learn personality traits. We hypothesize that this method can effectively capture the dynamic nature of human emotions and personality traits, as well as offer new insights into the mechanisms underlying these psychological constructs. To test our hypothesis, we conducted a study where participants watched a series of images and videos that were designed to elicit specific emotions. Our findings demonstrate that our CNN and LSTM networks were able to accurately extract both emotions and personality traits from the EEG data, and the sequence of stimuli helped to enhance the LSTM network’s ability to learn personality traits. The results showed that the combined CNN–LSTM approach outperforms existing methods and achieves high accuracy in classifying EEG signals into different emotional states. Therefore, the findings suggest that this approach could be leveraged to improve the personalization of emotion-recognition processes for users, enhancing the accuracy and efficacy of these systems. Additionally, this method may hold promise for facilitating emotional communication between humans and machines, enabling more natural and effective human–robot interaction.

The paper begins by providing an overview of the dataset used in the study and the pre-processing steps employed to refine the raw EEG signals. The feature-extraction techniques used to extract relevant features from the pre-processed EEG signals are then detailed. Subsequently, the CNN and LSTM networks utilized in the proposed approach are introduced, and the method in which they are combined to capture both spatial and temporal dependencies, in the EEG signals and individual differences in personality traits, is explained. In the results section, the performance of the proposed approach is presented and evaluated against existing state-of-the-art approaches using various evaluation metrics. The discussion section interprets the study results, emphasizing the importance of accounting for individual differences in personality traits in EEG-based emotion recognition. Finally, the study’s limitations are discussed, and recommendations for future research directions in this field are provided.

## 2. Materials and Methods

In this study, we applied a deep learning approach to analyze EEG data for emotion and personality trait classification. Specifically, we used a pre-trained CNN model to extract features from EEG data for emotion classification, as well as an LSTM network to predict personality traits based on the Big Five personality model. We also used a sequence of emotional stimuli to facilitate personality trait learning. In this section, we will describe the detailed procedures and materials used in the study, including the EEG recording, stimuli presentation, data pre-processing, and deep learning models.

### 2.1. Participants 

The study involved 270 healthy volunteers aged 19–30 years (mean 25.01, SD 3.13). There were three groups selected among volunteers, based on the highest score obtained by the IPIP-BFM-50, a commonly used scale of personality assessment. Along the way, the 50 items are rated on a 5-point scale depending on how faithful they are. The rating scale ranges between 1 (disagree), 3 (neutral), and 5 (agree). The test typically lasts 3–8 min [2]. The subjects were divided into Unstable Extrovert, Unstable Introvert, and Normal groups. More emotional stimulation was needed for stable people, so they were excluded from the experiment. Each group consisted of 20 healthy men (10 individuals) and women (10 individuals). The Unstable Introvert group is more likely to be depressed between the ages of 17 and 28 (mean age 25, SD 3.11). The Unstable Extrovert group is more likely to be manic from the ages of 18 to 29 (mean 25.21 and SD 3.07), and the Normal group is between 17 and 29 years of age (mean 26.32 and SD 3.12). The results of screening IPIP-BFM-50 participants are shown in Table 1. Men make up 48% of the population and women make up 52%, resulting in a balanced data set representative of the study population [7]. Table 1 provides observed IPIP-BFM-50 scales for 270 participants, along with their corresponding average and standard deviation values. The scales investigated in this study encompass five key personality traits. The table displays the standard deviations and average scores for each trait, reflecting the variability and central tendency within the participant group. As shown in the table, the personalities of participants are well balanced.

### 2.2. Emotional Stimuli

Based on the Russell model in Figure 1, emotions can be mapped into arousal and valence dimensions. Among several types of emotional stimulation, such as music, text, voice, we chose picture and movie because of its desirable properties, which were mentioned earlier [22]. Pictures were selected from the Geneva Affective Picture Database with the maximum arousal and valence scores for mentioned emotions. The stimuli used in this study consisted of 90 images and 3 videos, which were selected from 270 images and 12 videos, according to the maximum scores recorded by the Positive and Negative Affective Scheduling (PANAS) [28]. To evaluate the stimuli, 40 volunteers who did not participate in the main experiment were asked to rate them using the PANAS scale. Based on the arousal–valence scale of the Russell model, and in accordance with the personality groups, emotional stimuli were selected in sadness, happiness, and normal categories. The selection of stimuli was based on their potential to evoke distinct emotional responses. Detailed average and standard deviation scores for both images and videos, as assessed by PANAS, are presented in Table 2 and Table 3, respectively.

Table 2 presents the average and standard deviation scores for emotional image stimuli, as measured by the PANAS. The table displays the standard deviations, average scores, maximum values, minimum values, and the corresponding emotional stimulation for each category. The results show that the average scores for happiness, neutral, and sadness were 4.74, 4.31, and 4.52, respectively, with standard deviations of 1.25, 1.84, and 1.97. The maximum and minimum values for each category indicate the range of emotional intensity observed in the stimuli. Image stimuli were selected based on maximum PANAS scores in each category. 

Table 3 showcases the average and standard deviation scores for emotional video stimuli measured using the PANAS. The table provides insights into the emotional responses of the participants, including the scores for happiness, neutral, and sadness stimuli, along with their corresponding standard deviations, maximum values, and minimum values. Movie stimuli were selected based on maximum PANAS scores in each category.

### 2.3. Experiment Procedure

Throughout the experimental procedure, the participants were exposed to a diverse array of emotional stimuli comprising three categories—happy, sad, and neutral stimuli—in both images and videos. The participants were presented with a sequence of stimuli, with each sequence consisting of 30 images and a video of the same emotional valence at the end. EEG signals were recorded, while participants viewed stimuli, using a 32-lead EEG system with a standard 10–20 system. The impedance of each electrode was checked below 5 k-ohms. The set of 2-min eyes-open and 2-min eyes-closed EEG was recorded to calculate resting EEG features. All participants then completed the three types of emotion induction trials. In order to get noise--free data, the subjects were asked to keep control of their movement during the experiment. EEG signals were recorded using a Mitsar EEG Amplifier, with a sampling rate at 256 Hz. EEG recordings were taken in a quiet room, and participants were asked to sit comfortably in a chair and view images and videos without outside influences. The sequencing of these stimuli (happy, sad, and neutral) was conducted in a randomized fashion for each individual to avoid any potential biases or confounding variables. Specifically, each participant viewed 30 images and 1 video for each of the 3 emotional categories, resulting in a total of 90 images and 3 videos per participant (shown in Figure 2). Based on our previous study [29], and in order to investigate the potential impact of emotional stimuli on personality traits, the stimuli used in this study were sorted based on their arousal–valence score, ranging from the lowest to the highest, in order to accumulate the emotional effects of each stimulus and investigate potential differences in personality traits. By manipulating the emotional content of the stimuli, we aimed to elicit varied emotional responses in participants.

### 2.4. Pre-Processing

EEG signals recorded from 60 participants were pre-processed using a series of procedures and methods, as depicted in Figure 3. The signal was recorded at a sampling frequency of 256 Hz and, then, down sampled to a frequency of 128 Hz. The raw EEG were filtered using a Butterworth order 2 band-pass filter (0.4 Hz to 60 Hz) to remove high frequency noise, followed by a notch filter to remove power line noise (50 Hz). Then, the signal was visually inspected to remove inherent artifacts—such as eye blinks (EOG) and muscle activity (EMG)—that can skew the analysis. Finally, a spatial filter, based on Independent Component Analysis (ICA), was applied to remove the remaining noise sources and artifacts. The signal was divided into 2 s windows (hamming) with 50% overlap, corresponding to the duration of the presented stimuli for further analysis. Each window was baseline adjusted with the average amplitude of its first 200 ms [30]. 

### 2.5. Proposed Deep Learning Network

Deep learning is a field of machine learning that involves developing algorithms designed to learn the functionality of the human brain. Examples of deep learning algorithms include Convolutional Neural Networks (CNNs), Deep Neural Networks (DNNs), and Recurrent Neural Networks (RNNs). The term “deep” refers to the number of network layers employed, with deep networks containing over 100 layers compared to the 2 to 3 layers typically found in traditional networks. In this study, we proposed a conceptual model to learn features, recognize, and classify emotional valence and arousal labels, based on Russell model [31] of the presented EEG signals. A CNN network is employed to extract emotional features from the EEG signals, and a Long Short-Term Memory (LSTM) network is used to obtain personality traits from the extracted features without the explicit hand-crafted feature-extraction process. CNNs are commonly used in image processing, and they can learn to extract meaningful features from images. LSTM networks, on the other hand, are commonly used in sequence modeling, and they can learn to capture temporal dependencies in sequential data [32].

The conceptual model in Figure 4 illustrates the flow of the proposed approach, starting with the feature extraction from pre-processed EEG signals and ending with the predicted arousal and valence labels. The pre-processed EEG signals are fed into a CNN network, which utilizes spatial features for emotion classification. Furthermore, the spatial features obtained by the CNN network are used as inputs for the LSTM network, which focuses on capturing the temporal dynamics between these features for personality trait classification. The inputs of the CNN network are the extracted features from the EEG signal, and its outputs are the arousal and valence values of emotional stimuli. The inputs of the LSTM network are the features obtained from the flattening layer of the CNN, and its outputs are the values obtained for individual personality traits. The outputs of the CNN and LSTM networks are concatenated and passed through fully connected layers to shape the final stage emotion classifier. 

The CNN network consists of multiple convolutional layers followed by pooling layers, which capture local patterns in the EEG signals and reduce the dimensionality of the feature maps. The output of the CNN branch is a set of spatial features that captures the local patterns in the EEG signals. The second branch of the network is a LSTM network, which extracts personality traits from the CNN feature maps based on the Big Five personality traits (extroversion, agreeableness, conscientiousness, neuroticism, and openness) model. The LSTM network consists of multiple LSTM cells that capture the temporal dependencies in the EEG signals over time. The resulting model captures both the spatial and temporal dependencies in the EEG signals and the individual differences in personality traits, enabling accurate and robust emotion recognition. Several common emotion classifiers were investigated and compared, and based on the results, SVM was chosen as the final stage classifier for the proposed network. To optimize the hyperparameters of the SVM and prevent over-fitting, a nested cross-validation approach was employed.

### 2.6. Convolutional Neural Network(CNN)

In order to extract features from the EEG Signal for emotion classification, we utilized a pre-trained Convolutional Neural Network (CNN) [33]. Specifically, we chose VGG16, which is a widely used CNN architecture known for its high accuracy and robustness [34]. Before feeding the EEG Signal into the CNN, we first pre-processed the data, then segmented the continuous EEG signal into windows of 2 s, with a 50% overlap, resulting in windows per stimuli (image). Each window was converted into a 2D image by transforming the EEG time-series data into a spectrogram using the Short-Time Fourier Transform (STFT), with a window size of 512 samples and a step size of 256 samples. This resulted in a matrix of size 128 × 129, where the frequency dimension ranged from 0.4 Hz to 60 Hz, and the time dimension ranged from 0 to 2 s. The Short-Time Fourier Transform (STFT) of a signal $x\left(t\right)$ is defined as:(1)$$STFT\left(\tau ,\omega \right)=\int_{-\infty }^{+\infty }x\left(t\right)h(t-\tau )e^{-j\omega t}dt$$
where h(t) is a window function, τ is the window position on the time axis, and ω is the frequency variable. We then fed the pre-processed EEG data into the VGG-16 CNN and used the output from the last convolutional layer (conv5_3) as the feature representation. This layer has been shown to capture high-level features and patterns from visual stimuli, which we hypothesized would also be useful for extracting features from EEG data. 

The VGG-16 CNN (Convolutional Neural Network) is a deep learning architecture developed by the Visual Geometry Group at the University of Oxford [35]. It consists of 16 layers, including 13 convolutional layers and 3 fully connected layers. The architecture of the VGG-16 CNN is characterized by its use of small filters (3 × 3) with a small stride size (1 × 1), which allows for a more fine-grained analysis of the input data. The convolutional layers are followed by max-pooling layers, which reduce the spatial dimensions of the feature maps and increase their translation invariance [34,36]. Our approach of using a pre-trained CNN to extract features from EEG data allowed us to effectively capture complex patterns and representations that are difficult to extract using traditional signal-processing techniques. The resulting feature representation can be used for further analysis and the classification of emotional states (Figure 5).

The steps of the VGG-16 CNN are:The input image is passed through 13 convolutional layers, which utilize 3 × 3 receptive field filters, with a 1 × 1 convolution filter for linear transformation of the input.After each convolutional layer, a Rectified Linear Unit (ReLU) activation function is applied to introduce non-linearity in the network, which is a linear function that generates zero output for negative inputs and a matching output for positive inputs.Max-pooling layers are used to reduce the dimensionality and number of parameters in the feature maps generated by each convolution step. This step is necessary due to the increase in the number of available filters from 64 to 128, 256, and 512 in the final layers.The output from the last convolutional layer is flattened and fed into three fully connected layers.Each fully connected layer is followed by a ReLU activation function, except for the last one, which uses a SoftMax activation function to output class probabilities.During training, the weights of the network are updated using backpropagation with stochastic gradient descent.

### 2.7. Long Short-Term Memory (LSTM) Network

In our study, we used an LSTM (Long Short-Term Memory) network to extract personality traits from the pre-processed EEG data. The LSTM network is a type of recurrent neural network (RNN) that is well-suited for modeling sequential data, such as time-series data. The key advantage of LSTM networks over traditional RNNs is that they are able to capture long-term dependencies in the data by using a memory cell that can store information over time [11]. The architecture of our LSTM network, in Figure 6, consisted of three LSTM layers, each followed by a fully connected layer. The LSTM layers were responsible for capturing the temporal dynamics of the EEG data and learning the relationships between different time points. The fully connected layers were used to map the output of the LSTM layers to the Big Five personality traits (extroversion, agreeableness, conscientiousness, neuroticism, and openness). We used flattening layer features, extracted from the pre-processed EEG data by CNN, as inputs to the LSTM network. Before proceeding with the fully connected linked layer in CNN, it is necessary to use the flattening layer to create a one-dimensional vector as input for LSTM. The EEG data were divided into windows of fixed duration, and each window was labeled with the participant’s personality traits. We used mean squared error as the loss function to train the LSTM network, which measures the difference between the predicted and actual personality trait values. We also used the Adam optimizer, which is a stochastic gradient descent optimizer that adapts the learning rate based on the gradient of the loss function.

To evaluate the performance of the LSTM network, we used a leave-one-subject-out cross-validation approach, where we trained the LSTM network on all but one participant’s data and tested its performance on the left-out participant’s data. By using an LSTM network to model the temporal dynamics of the EEG data, we aimed to improve the accuracy of personality trait prediction compared to common classifier networks that do not take into account the sequential nature of the data [14].

The number of neurons in each LSTM layer decreases as you move deeper into the network. Deeper layers tend to capture more abstract features and, therefore, require fewer neurons to represent the information effectively. Therefore, the first two hidden layers contain 64 and 32 neurons, respectively, and use ReLU as an activation function to learn nonlinear representations. The third LSTM layer contains 16 neurons and uses a sigmoid activation function. The dense layer also uses a sigmoid activation function. A sigmoid activation function is useful for the output layer of the network when dealing with binary classification problems or tasks that require the model to output probabilities. The LSTM and dropout layers are used to learn features from EEG signals. The output of the fully connected layers, to predict the participant’s Big Five personality traits (extroversion, agreeableness, conscientiousness, neuroticism, and openness) and the dense layer, is used for classification [27].

### 2.8. Implementing and Setting the Hyperparameters

The network was implemented and trained using the Python programming language version 3.7. The Keras library, which is a high-level neural network API written in Python and capable of running on top of TensorFlow, was used to develop the network architecture. The TensorFlow backend was used for efficient numerical computations on GPUs, which was important for speeding up the training process. The hardware used for training and testing the network was a computer with an Intel Core i7 processor, 16 GB of RAM, and an NVIDIA GeForce GTX 1080 Ti graphics card. The operating system used was Ubuntu 18.04. In addition, the NumPy and Pandas libraries were used for data pre-processing and analysis. The code was written and executed in Jupyter Notebook, an open-source web application that allows for creating and sharing documents containing live code, equations, and visualizations. In training the proposed network, several hyperparameters were specified. The learning rate was set to 0.001, which determines the step size at each iteration while updating the parameters of the network. The batch size was set to 64, which specifies the number of samples used in each iteration for the gradient descent. The number of epochs was set to 500, which determines the number of times the entire training set was passed through the network during training. The optimizer that was used was Adam, which is a widely used stochastic optimization algorithm that uses adaptive learning rates to update the network weights. These hyperparameters were chosen based on previous research and experimentation, and they were adjusted during training, using the validation set, to achieve the best possible performance of the network.

### 2.9. Classifier Evaluation

To evaluate the performance of the proposed CNN–LSTM network, we used the following evaluation metrics:1.Accuracy: The classifier accuracy is generally calculated from Equation (2):
(2)$$Accuracy=\frac{\left(TN+TP\right)}{\left(TN+FN+TP+FP\right)}$$2.Precision: The following equations are used to evaluate the precision of the classifier of emotion classes in Equation (3).
(3)$$Precision=\frac{TP}{TP+FP}$$3.Recall: It is the proportion of correctly classified positive emotions to the total number of actual positive emotions. It is calculated as follows:
(4)$$Recall=\frac{TP}{TP+FN}$$4.F1 Score: It is the harmonic mean of precision and recall, which is obtained according to Equation (5).
(5)$$F1score=\frac{2\times (Precision\times Recall)}{Precision+Recall}$$
where TP (true positive) represents the number of correctly classified positive emotions, TN (true negative) represents the number of correctly classified negative emotions, FP (false positive) represents the number of falsely classified positive emotions, and FN (false negative) represents the number of falsely classified negative emotions.

To assess the performance of the proposed CNN–LSTM network, we also used the confusion matrix. The confusion matrix is a table that summarizes the performance of the classifier by showing the number of true positives, true negatives, false positives, and false negatives for each class. In addition, we used the Receiver Operating Characteristic (ROC) curve to plot the true positive rate (TPR) against the false positive rate (FPR) at various threshold levels and the Area under the Curve (AUC) metric to evaluate the performance of the proposed deep learning network. The ROC curve is a graphical plot that illustrates the performance of the classifier as its discrimination threshold is varied. The AUC is a metric that measures the overall performance of the classifier, taking into account all possible discrimination thresholds. The AUC score ranges from 0 to 1, with 1 indicating perfect classification performance and 0.5 indicating random chance. All these evaluation metrics will be used to assess the performance of the proposed network in emotion recognition using EEG signals.

The data were split into training, validation, and testing sets using a 70–15–15 ratio, respectively. This means that 70% of the data were used for training, 15% were for validation, and 15% were for testing. The splitting process was performed randomly, ensuring that the data from each group (Unstable Introvert, Unstable Extrovert, and Normal) were equally represented in each set. The training set was used to train the proposed network, while the validation set was used to adjust the hyperparameters and prevent overfitting. The testing set was used to evaluate the performance of the trained network on unseen data. This process of data splitting helped to ensure that the network was trained and evaluated on a diverse set of data and that it can generalize well for new data.

## 3. Results

In this study, the performance of the proposed deep learning network, with the influence of personality traits in classifying emotions based on EEG signals, was evaluated using various metrics, including accuracy, loss, precision, recall, and F1-score. In order to evaluate the overall performance of the network, Receiver Operating Characteristic (ROC) curves and Area under the Curve (AUC) scores were utilized. The ROC curve is a graphical representation of the trade-off between the true positive rate (TPR) and the false positive rate (FPR) of a binary classifier system, where the AUC score represents the overall performance of the classifier.

### 3.1. The Convolutional Neural Network VGG-16

The CNN–VGG-16 network was found to be highly effective in extracting relevant EEG features related to emotion states in arousal and valence dimensions. The network was trained on the dataset of EEG recordings and achieved an accuracy of 86.3% on the training set and 87.30% on the testing set. This suggests that the CNN–VGG-16 network was able to learn and generalize patterns related to emotion states from the EEG data. The accuracy, precision, recall, and F1-score of the CNNs are summarized in Table 4. The overall accuracy achieved by the network was 87.30%, with precision, recall, and F1-score measures of 92.90%, 92.67%, and 92.78% for the arousal and valence dimension. These metrics were obtained from the TP (true positive), TN (true negative), FP (false positive), and FN (false negative) parameters generated by testing the network that is illustrated in Table 5. 

In our study, the CNN–VGG-16 network achieved AUC scores of 0.79%, 0.84%, 0.83%, and 0.80% for four emotion classes—high arousal-high valence (HV-HA), low arousal-high valence (LV-HA), high arousal-low valence (HV-LA), and low arousal-low valence (LV-LA)—respectively. For example, to evaluate the performance of the classifier for the high arousal-high valence class, we trained a binary classifier to distinguish between high arousal-high valence and all other classes. We then computed the true positive rate (TPR) and false positive rate (FPR) for this classifier at different thresholds and plotted these values on an ROC curve. We repeated this process for each of the other three classes. A perfect classifier would have an AUC score of 1, while a random classifier would have an AUC score of 0.5 (given as a dotted line on the ROC plots shown in Figure 7). The ROC curves for each emotion class are presented in Figure 7a–d, which demonstrate the ability of the CNN to correctly classify the emotional states, as evidenced by the high TPR and low FPR values. These results suggest that the CNN network was able to effectively distinguish between different emotional states in both arousal and valence dimensions.

### 3.2. The LSTM Network

In this study, the LSTM network was trained to extract five personality traits (extroversion, openness, neuroticism, agreeableness, and conscientiousness) from the CNN flattening layer features. The network achieved an accuracy of 91.29% on the training set and 90.72% on the testing set, indicating that the LSTM network was also effective in learning patterns related to personality traits from the EEG data. Table 6 shows the performance measures of the LSTM model used in this study. The table presents the accuracy, precision, and F1-score of the model’s performance in predicting five personality traits. The performance measures, including accuracy, precision, and F1-score are calculated by using TP, TN, FP, and FN values that are taken from the confusion matrix. The results reveal that the model has achieved high accuracy scores across all personality traits, with the highest accuracy being 92.54% in neuroticism and the lowest being 89.12% in agreeableness. In terms of precision, the model has attained precision rates above 95% for all personality traits, with the highest precision being 96.79% in extroversion and the lowest being 95.78% in openness. The F1-scores, which take into account both precision and recall, also demonstrate high scores, with the highest being 95.84% in neuroticism and the lowest being 93.95% in agreeableness.

### 3.3. Performance of the Proposed Network

The present study utilized a CNN–VGG-16 network to extract EEG features, based on emotion states in the arousal and valence dimensions, as well as a LSTM network to extract five personality traits. The extracted features from both networks were then used as inputs to a classifier for emotion recognition in Figure 8. In order to compare the accuracy of the proposed deep learning network with LSTM, in contrast with common emotion classifiers, the performances of several widely used classifiers [37] in EEG-based emotion recognition, including Naive Bayesian (BN), Support Vector Machines (SVM), and K-nearest neighbor (KNN), were assessed. For this purpose, the extracted features from the flattening layer of the CNN network were fed as input to these classifiers, and the outputs of the classifiers were chosen as the values of arousal and valence.

Figure 9 displays the comparative performance analysis of common emotion classifiers and the proposed network for all participants. Remarkably, the proposed network with LSTM exhibited accuracy rates with average values of 93.97%. In contrast, the KNN, SVM, and BN networks demonstrated average accuracy levels of 84.64%, 86.91%, and 84.88%, respectively. Notably, there exists a significant difference between SVM and other classifiers, as indicated by a p-value greater than 0.01. The precision and F1-score for the proposed network were 91.37% and 89.61%, respectively. The results demonstrate that the LSTM network was able to extract more information, in comparison with other classifiers, from the same features related to emotions. 

Figure 9 illustrates the loss and accuracy of the proposed network. As shown in Figure 9a, the loss and accuracy of the model converge after about 80 and 120 epochs, respectively. Within the first 40 epochs, model accuracy of 91.23% was achieved for low arousal-high valence (LV-HA). The accuracy improved gradually from epoch 40 to 140, reaching 92.12%. The model then stabilized between epochs 140 and 250, ultimately achieving a high accuracy of 94.68%. Figure 9b displays the convergence of the algorithm’s loss and accuracy, at roughly 30 and 40 epochs, respectively. The accuracy steadily increased from epoch 40 to 160, reaching a maximum of 92.63%. Following this, the model maintained stability between epochs 160 and 250, ultimately achieving a high accuracy level of 93.91%. Figure 9c, low arousal-low valence (LV-LA), demonstrates that the network’s loss and accuracy reach convergence after approximately 80 and 60 epochs, respectively. The accuracy displays gradual improvement from epoch 60 to 120, culminating in a peak of 94.21%. Between epochs 120 and 250, the model stabilizes, eventually attaining a high accuracy level of 94.83%. Finally, the algorithm’s loss and accuracy, presented in Figure 9d for high arousal-low valence (HV-LA), converge at around 40 and 40 epochs, respectively. The accuracy steadily improves from epoch 40 to 60, peaking at 92.75%, before stabilizing between epochs 60 and 250. Ultimately, the model achieves a high accuracy of 92.43%.

The results of the four commonly used classifiers, applied as the final stage classifier in the proposed network, are presented in Table 7. The SVM classifier outperforms the other classifiers, achieving the highest accuracy of 93.97%, precision of 91.37%, and F1 score of 89.61%. In contrast, the MLP, KNN, and BN classifiers achieved accuracies of 79.21%, 81.48%, and 82.71%, respectively, with lower precision and F1-scores. Therefore, based on these findings, we have selected the SVM as the best final stage classifier (Figure 4) for the proposed network, due to its superior performance.

### 3.4. Comparison of Computaional Time

Furthermore, we assess the computational efficiency of different machine learning models to determine their practical usability. We evaluate the training and inference times as indicators of computational efficiency. The specific training and inference times for each machine learning model are documented in Table 8. It is important to note that all experiments were conducted under consistent conditions within a controlled environment. We observed that the proposed network required relatively longer training and inference times (1889 s and $1.8\times 10^{-5}s$) compared to traditional machine learning models such as CNN–LSTM and CNN–SVM, which aligns with our expectations. However, the actual inference time remains low across all machine learning models. In the case of the proposed network, there exists a trade-off between the complexity of the model’s structure and its performance.

### 3.5. Comparision of Common Deep Learning Networks

Table 9 is a comparison for several recent emotion classification studies based on deep learning, especially CCN networks. As shown in Table 9, the proposed network achieved an accuracy of 93.97% on the original dataset, consisting of 90 images and 3 videos, with 60 participants and utilizing a 32-channel EEG. The CNN model, trained on the DEAP dataset of 40 music videos, achieved an accuracy of 82.24% using a combination of 32-channel EEG, ECG, and GSR signals. The CNN–LSTM model, also trained on the DEAP dataset, achieved an accuracy of 88.87% using 32-channel EEG. The CNN–SAE model, trained on the SEED dataset of 15 music videos, achieved an accuracy of 96.77% utilizing a 62-channel EEG. The results show that, among the common models, the proposed network demonstrated the accuracy of 93.97% on the original dataset, which included relatively small-sized training data. This suggests that the proposed network can achieve acceptable accuracy levels even with limited training data. Additionally, the proposed network utilized a 32-channel EEG signal, making it a feasible and practical choice for emotion recognition tasks. These findings highlight the effectiveness of the proposed network in accurately recognizing emotions from EEG signals, showcasing its potential for real-world applications.

## 4. Discussion

Emotion recognition using biological signals has become popular due to its high temporal resolution and cost-effectiveness. Emotions are influenced by various factors, including personality traits. Individuals with different personality traits exhibit different responses to a distinct stimulus. In most studies, personality traits are not quantitatively incorporated into the emotion recognition network. In this study, we attempted to identify personality traits using an LSTM network and quantitatively incorporate these traits into the new emotion recognition network. The results are promising, as they show that incorporating personality traits’ information into the emotion classifier network has led to an increase in the overall accuracy of the emotion classifier.

Furthermore, we demonstrated the effectiveness of using a sequence of emotional stimuli to facilitate personality trait learning. Our results showed that the LSTM network achieved higher accuracy in predicting personality traits when the participants watched a sequence of emotional stimuli compared to when they watched a single type of stimuli. This finding suggests that the sequence of emotional stimuli can enhance the representation learning of personality traits from EEG data. Additionally, our study showed that the proposed stimulation enables the network to achieve high accuracy with a smaller input data size compared to a common emotion dataset, such as DEAP.

Finally, as the limitations of this study, we can mention that accurately measuring induced emotions during video watching is not feasible. In this study, we assumed a consistent level of emotional valence is induced in individuals throughout the video, which may not be entirely realistic. For future studies, it is recommended to continuously monitor an individual’s emotional state during an emotional video by using a keypad and utilizing the observed valence level as output for network training. It is hoped the networks can be trained more effectively, even with a smaller dataset size. Future works could include the following areas of exploration:Robustness and Generalizability: Evaluate the performance and generalizability of the proposed network and the selected SVM classifier on larger and diverse datasets. Assess the model’s ability to handle different scenarios, such as varying input data sizes, different EEG channels, and diverse populations. This analysis can provide insights into the model’s robustness and its potential for real-world applications.Incremental Learning: Investigate methods to enable the proposed network to adapt and learn, continuously, as new data become available. Explore techniques such as online learning or incremental learning to update the model over time and accommodate changes in the data distribution or concept drift. This would enhance the network’s adaptability and long-term performance in dynamic environments.Integration of Multimodal Data: Emotion recognition can benefit from the integration of multiple modalities, such as facial expressions, physiological signals (e.g., ECG, GSR), or audio features. Investigate the integration of EEG data with other modalities to enhance the accuracy and robustness of emotion classification. Explore fusion techniques and feature extraction approaches to effectively combine information from different sources.Real-Time Implementation: Implement the proposed network and the selected SVM classifier in real-time applications or wearable devices. Evaluate the feasibility of deploying the model in resource-constrained environments and assess its real-time performance. Consider the trade-off between accuracy and computational requirements, aiming to strike a balance that meets the real-time constraints of practical applications.

## 5. Conclusions

In this study, we proposed a novel approach for emotion classification, using EEG data and deep learning techniques, while considering personality traits. Our approach leverages the power of pre-trained CNNs and LSTMs to extract high-level features from EEG data and predict personality traits based on the Big Five personality model. In the proposed model, the LSTM network is fed by extracted features of the flattening layer of the CNN–VGG-16. Our results showed that the proposed approach achieved high accuracy in classifying and predicting personality traits. Specifically, the LSTM model achieved an average accuracy of 90.72% in predicting the personality traits, while the CNN model achieved an accuracy of 87.30% in classifying emotions. By combining the trained CNN as the emotion classifier and a trained LSTM as the personality traits classifier, a new classifier network has been created that has the ability to classify emotions with higher accuracy. Our results show that the overall accuracy of the proposed network for emotion recognition, by considering personality traits, was 93.97%. For more comparisons, the performance of several recent emotion classification studies, with a focus on CNN and LSTM networks, were investigated. The proposed network demonstrated competitive accuracy levels despite the relatively small-sized training data.

## Figures and Tables

**Figure 1 brainsci-13-00947-f001:**
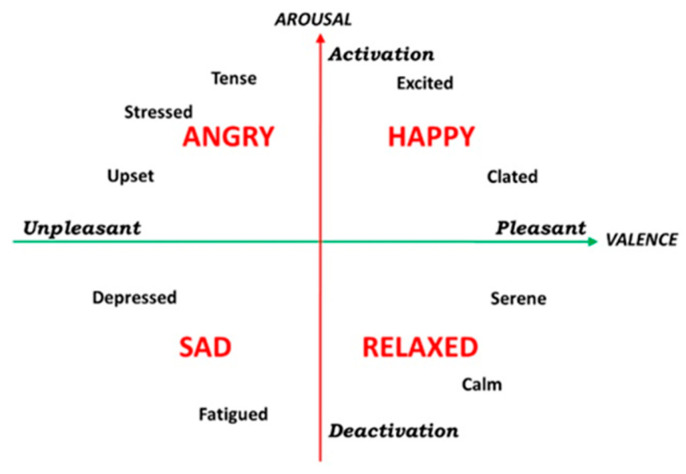
According to the Russell Emotional Model, emotions can be expressed quantitatively in two dimensions: arousal and valence. As shown, sadness and happiness are the two most different points.

**Figure 2 brainsci-13-00947-f002:**
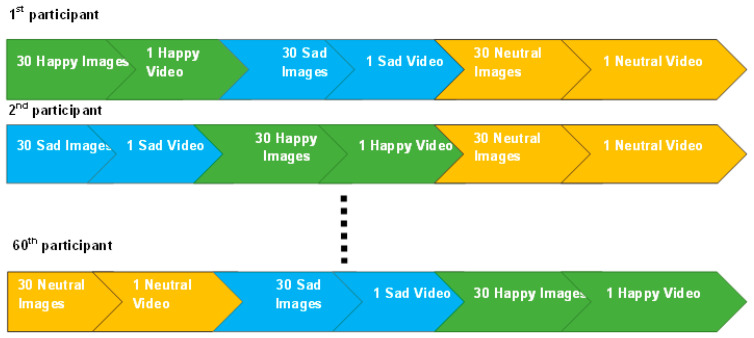
The emotional stimulus sequence presented to participants in the study, consisting of 90 images and 3 videos. The order of stimuli was randomized for different participants.

**Figure 3 brainsci-13-00947-f003:**
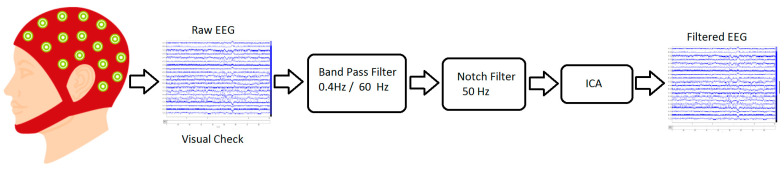
The block diagram of the EEG Pre-processing procedure.

**Figure 4 brainsci-13-00947-f004:**
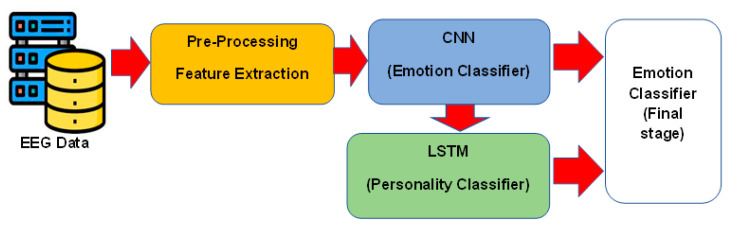
The block diagram of the proposed deep learning network.

**Figure 5 brainsci-13-00947-f005:**
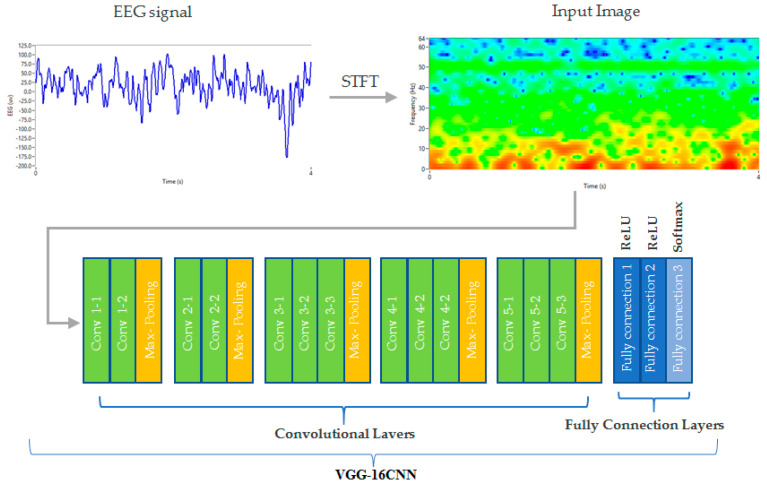
The schematic of the VGG16 (CNN) used in the study. The STFT of the EEG is fed into the CNN as an input feature.

**Figure 6 brainsci-13-00947-f006:**
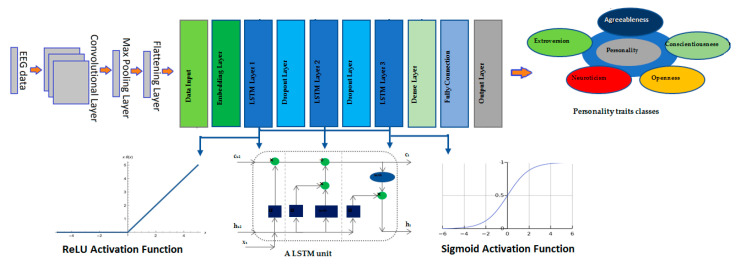
The architecture of the LSTM network used in the study.

**Figure 7 brainsci-13-00947-f007:**
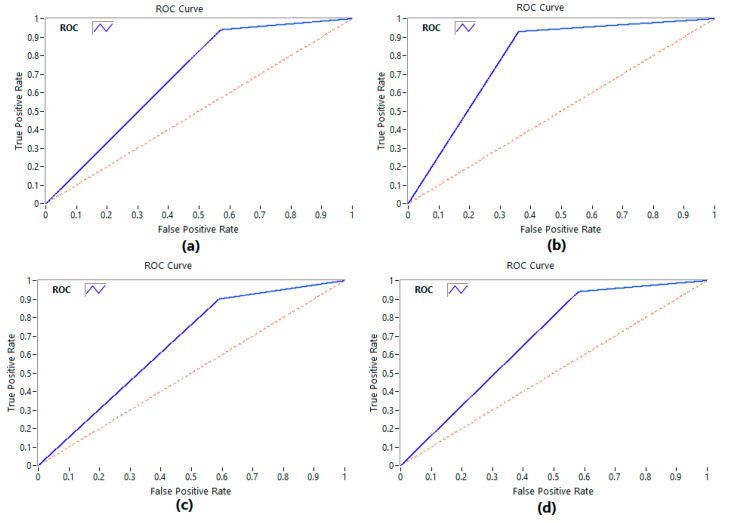
ROC curve of a CNN based on VGG-16: (**a**) low arousal-high valence (LV-HA), (**b**) high arousal-high valence (HV-HA), (**c**) low arousal-low valence (LV-LA), and (**d**) high arousal-low valence (HV-LA).

**Figure 8 brainsci-13-00947-f008:**
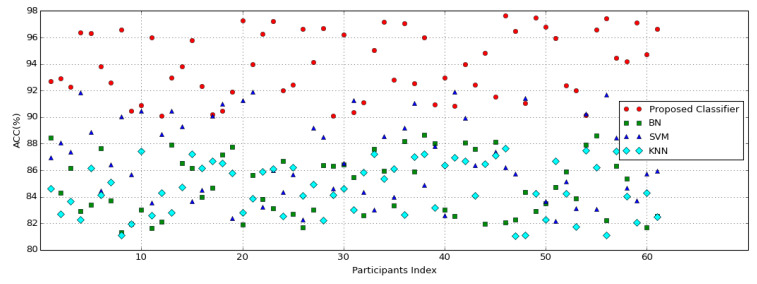
The average accuracy of KNN, SVM, and BN classifiers, as well as the proposed network for all participants.

**Figure 9 brainsci-13-00947-f009:**
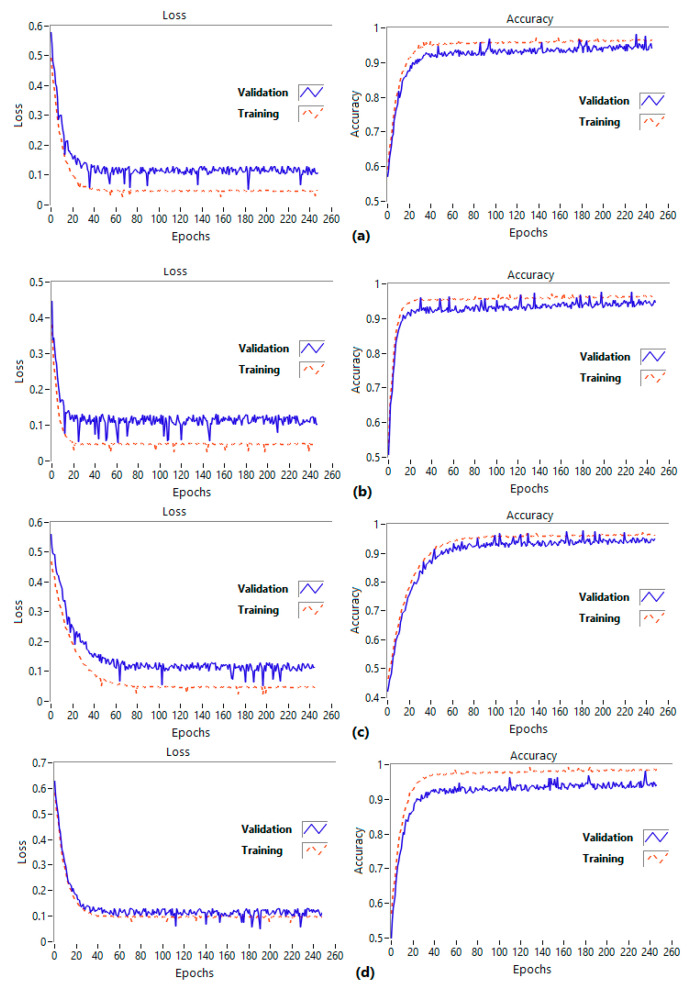
The loss and accuracy of the proposed network for (**a**) low arousal-high valence (LV-HA), (**b**) high arousal-high valence (HV-HA), (**c**) low arousal-low valence (LV-LA), and (**d**) high arousal-low valence (HV-LA).

**Table 1 brainsci-13-00947-t001:** Average and standard deviation of IPIP-BFM-50 scales obtained by all participants.

STD					AVERAGE					PARTICIPANTS		
E	A	C	N	O	E	A	C	N	O	QTY	age	sex
1.1	1.6	1.65	1.21	1.34	2.21	1.98	2.41	2.68	2.71	141	19–30	F
1.1	1.7	1.82	1.53	1.25	2.45	2.33	1.95	1.96	2.37	129	19–30	M

(E: Extroversion, A: Agreeableness, C: Conscientiousness, N: Neuroticism, and O: Openness).

**Table 2 brainsci-13-00947-t002:** The average and standard deviation scores for the emotional image stimuli measured by PANAS.

STD	AVERAGE	MAX	MIN	Stimulation
1.05	3.74	7	1	Happiness
0.94	3.31	7	1	Neutral
1.17	3.52	7	1	Sadness

**Table 3 brainsci-13-00947-t003:** The average and standard deviation scores for the emotional video stimuli measured by PANAS.

STD	AVERAGE	MAX	MIN	Stimulation
0.82	3.38	7	1	Happiness
0.91	3.66	7	1	Neutral
1.08	2. 72	7	1	Sadness

**Table 4 brainsci-13-00947-t004:** Classification performances for the CNN-VGG-16.

Emotion Dimension	Accuracy (%)	Precision (%)	Recall (%)	F1-Score (%)
HV-HA	89.77	95.16	93.16	94.15
HV-LA	86.70	91.30	93.56	92.42
LV-HA	88.65	93.31	94.05	93.68
LV-LA	84.07	91.85	89.92	90.87

**Table 5 brainsci-13-00947-t005:** The percentage of TP (true positive), TN (true negative), FP (false positive), and FN (false negative) parameters in the CNN–VGG-16 network.

	Emotion Classes			
	HV-HA	HV-LA	LV-HA	LV-LA
TP	17.7%	18.9%	23.7%	21.4%
TN	1.6%	1.3%	1.3%	1.3%
FP	0.9%	1.8%	1.7%	1.9%
FN	1.3%	1.3%	1.5%	2.4%

**Table 6 brainsci-13-00947-t006:** Performance of the LSTM network for personality traits.

Dimension	Accuracy (%)	Precision (%)	F1-Score (%)
Extroversion	90.28	96.79	94.52
Openness	91.76	95.78	95.50
Neuroticism	92.54	96.65	95.84
Agreeableness	89.12	96.45	93.95
Conscientiousness	89.90	96.11	94.28

**Table 7 brainsci-13-00947-t007:** The accuracy, precision, and F1-score for different final stage classifiers.

Classifier	Accuracy (%)	Precision (%)	F1-Score (%)
SVM	93.97	91.37	89.61
MLP	79.21	78.53	75.96
KNN	81.48	79.94	76.42
BN	82.71	80.33	80.29

**Table 8 brainsci-13-00947-t008:** The computational time and performance for the proposed network and common deep learning networks.

Model	Accuracy (%)	Precision (%)	F1-Score (%)	Training Time (second)	Inference Time (second)
Proposed Network	93.97	91.37	89.61	1889	$1.8\times 10^{-5}$
CNN-LSTM	87.34	85.29	82.98	1325	$1.6\times 10^{-5}$
CNN-SVM	86.91%	83.75%	81.15%	987	$0.9\times 10^{-5}$

**Table 9 brainsci-13-00947-t009:** A comparison of the proposed network with common deep learning emotion classifiers.

Model	Data Set	Stimuli Size	Number of Participant	Electrophysiological Signal	Accuracy (%)
Proposed Network	original	90 Images and 3 videos (One minute)	60	32 channels EEG	93.97
Multi-Column- CNN [38]	DEAP	40 music videos (One minute)	32	32 channels EEG	90.1%
CNN-LSTM [25]	DEAP	40 music videos (One minute)	32	32 channels EEG	88.87%
CNN-SAE [39]	SEED	15 music videos (4 min)	15	62 channels EEG	96.77%

## Data Availability

The data presented in this study are available on request from the corresponding author.

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
