# Peer review of "Personality-Based Emotion Recognition Using EEG Signals with a CNN-LSTM Network"

_brainsci, 2023, doi:10.3390/brainsci13060947_

Round 1

Reviewer 1 Report

Comments and Suggestions for Authors

The article proposes a novel approach for emotion recognition using EEG signals and the Big Five personality traits. The approach combines a pre-trained CNN to extract emotion-related features from the raw EEG data and an LSTM network to extract features related to the Big Five personality traits. The study used 30 images and one video selected from the Positive and Negative Affect Schedule (PANAS) to elicit different emotional states in participants. The EEG signals recorded from participants were pre-processed using a series of steps and methods to prepare them for input into a CNN-LSTM network. The proposed approach achieved high accuracy in classifying emotions and predicting personality traits, outperforming baseline methods. The study demonstrates the potential of using deep learning techniques for EEG-based emotion and personality analysis.

Strengths: - The article presents a novel approach for emotion recognition using EEG signals that combines CNNs and LSTM networks. - The study conducted EEG recordings from volunteer participants and compared the performance of the proposed approach with existing state-of-the-art approaches for EEG-based emotion recognition. - The findings demonstrate that the CNN and LSTM networks were able to accurately extract both emotion and personality traits from the EEG data. - The results showed that the combined CNN-LSTM approach outperforms existing methods and achieves high accuracy in classifying EEG signals into different emotional states.

Weaknesses: - The article lacks clarity in explaining the methodology used for data collection and analysis. The authors do not provide enough information on the selection criteria for participants, the stimuli used to elicit emotions, and the preprocessing steps applied to the EEG data. - The article does not provide a detailed explanation of the CNN and LSTM architectures used in the study. The authors do not explain how the hyperparameters were selected, and how the models were trained and validated. - The article does not provide a thorough discussion of the limitations of the study. The authors do not address potential confounding factors that may affect the EEG signals, such as environmental noise, participant fatigue, and electrode placement. - The article lacks a clear contribution to the field of emotion recognition. The authors do not explain how their approach differs from existing methods, and how it can be applied in real-world scenarios.

Limitations: - The study was conducted on a small sample size of 270 healthy volunteers, which may not be representative of the general population. The authors do not explain how the sample size was determined, and whether the results can be generalized to other populations. - The study only focuses on the Big Five personality traits, which may not capture the full range of personality dimensions. The authors do not explain how their approach can be extended to other personality models. - The study only uses EEG signals to recognize emotions, which may not be sufficient to capture the complexity of human emotions. The authors do not explain how their approach can be combined with other modalities, such as facial expressions and physiological signals.

Conclusion: the article presents a promising approach for emotion recognition using EEG signals, but it lacks clarity in explaining the methodology and limitations of the study. The authors need to provide more details on the data collection and analysis, and address potential confounding factors that may affect the results.

To improve the article for publication in the Brain Sciences journal, the authors should consider the following suggestions: 1. Provide more details on the data collection and analysis, including the selection criteria for participants, the stimuli used to elicit emotions, and the preprocessing steps applied to the EEG data. 2. Provide a detailed explanation of the CNN and LSTM architectures used in the study, including how the hyperparameters were selected, and how the models were trained and validated. 3. Address potential confounding factors that may affect the EEG signals, such as environmental noise, participant fatigue, and electrode placement. 4. Provide a clear contribution to the field of emotion recognition, explaining how their approach differs from existing methods, and how it can be applied in real-world scenarios. 5. Discuss the limitations of the study in more detail, including the small sample size, the focus on the Big Five personality traits, and the use of only EEG signals to recognize emotions. 6. Provide recommendations for future research directions in this field. 7. Discuss and compare against relevant studies in the research field. Discuss the following topics - Four-classes human emotion recognition via entropy characteristic and random forest. - Deep convolutional neural Network‐Based visual stimuli classification using electroencephalography signals of healthy and Alzheimer’s disease subjects. - An efficient mixture model approach in brain-machine interface systems for extracting the psychological status of mentally impaired persons using EEG signals. - Recognition of human inner emotion based on two-stage FCA-ReliefF feature optimization.

Reviewer 2 Report

Comments and Suggestions for Authors

This work proposes an EEG-based emotion recognition method using a CNN-LSTM network and achieves a good accuracy of 93.97% by considering personality traits. Besides, a total of 60 participants were recruited, and their EEG data were recorded when viewing unique sequence stimuli designed to effectively capture the dynamic nature of personality traits. The objective of this work is clear, and the proposed technique sounds available. However, several critical issues should be further clarified and improved, as listed below:

1. The citations are improper in the manuscript, please cite with numbers on the corresponding places in the text.

2. The authors emphasized that EEG-based emotion recognition combines CNN and LSTM is one of the contributions in this work. However, to the best of my knowledge, the CNN-LSTM model has already been applied in several previous works, and those also considered individual characteristics, i.e., personality traits. Therefore, it is not clear to me why the authors claimed this model was new. In this regard, the authors should reorganize the motivations and highlight their significance with strong evidence, compared to other CNN-LSTM works. Otherwise, this is only repetitive work in this field.

3. Concerning the data pre-processing, the EEG signals were segmented by 2 seconds with a 50% overlap. Why use this setting? Is it the most proper? Meanwhile, what is the window employed? Different settings and window functions can generate various performances that impact the final results accordingly. Therefore, it is necessary to demonstrate the appropriateness of such operations.  

4. Regarding the experimental dataset, is it imbalanced or not? Moreover, the experimental stimuli include three states: happy, sad, and neutral. Then in the classification task, four classes are considered in terms of arousal and valence. So, how to connect discrete emotions (happy, sad, and neutral) with dimensional emotions (arousal and valence)? Besides, how about the ground truth?  

5. The authors should include the overall complexity and computation cost of the proposed method. Moreover, to facilitate reproducible research, I suggest that the authors release the source codes on github.com and provide the link. Also, it is meaningful to release the experimental dataset. Consequently, I strongly advise that the authors provide a manner to assess the source codes and experimental data, which would make a positive effect on the academic community.

6. The performance evaluation based on only one dataset can not significantly demonstrate the practical applicability of the proposed method, the cross-corpus results are vital in emotion recognition, as emotion generally exhibits individual differences. Hence, the results from other public emotional datasets, such as SEED, DEAP, DREAMER, GAMEEMO, etc. are also essential.  

7. How many kinds of features were applied in the proposed CNN+LSTM model? A table that summarizes the applied features extracted from the raw EEG data is better.  

8. As for the results, the performances vary in the four emotional dimensions (Table 1). Why HV-HA offers the best and LV-LA shows the worst? Can you further analyze the variations and provide the reasons? Besides, how to obtain the results using KNN, SVM, MLP, and BN? What are the input features and output categories of these conventional classifiers? The overall presentation of the results and analysis is not good enough. More clear descriptions of the testing results are needed. Meanwhile, the comparative histograms are suggested to be appended.  

9. A comparative study with the previous related works is missing in the discussion section. Comparisons should not only include the evaluated metrics, such as accuracy, precision, recall, and F1 Score, but also contain the data sources, time complexity, computation costs, and so on. Otherwise, it is difficult to demonstrate this work is state-of-the-art in this field.

10. A conclusion section is needed, which not only summarizes the advantages and improvements of the proposed method but also includes the limitations and shortcomings that can be enhanced in future works.

Comments on the Quality of English Language

The authors should spend time revising this manuscript. It is not easy to follow in the current form. Language is a part of the issue, but more importantly, the quality of the presentation needs to be worked on. I strongly suggest the authors read more high-quality papers in this field. Extensive editing of the English language is needed.    

Reviewer 3 Report

Comments and Suggestions for Authors

The paper at hand introduces an intelligent recognition system employing a fusion of a convolutional neural network (CNN) and a long short-term memory (LSTM) one. The aim is to classify EEG signals into 4 emotional states, i.e., Low Valence - High Arousal (LV-HA), Low Valence - Low Arousal (LV-LA), High Valence - Low Arousal (HV-LA) and High Valence - High Arousal (HV-HA). The exploited CNN architecture is the well-established VGG-16 and is responsible for extracting emotional features from the input signals, while the LSTM network captures the personality traits of the subject based on the OCEAN principles. To achieve that, a new emotional database including 60 participants is collected, annotated and carefully assessed. In general, the paper is novel and the results clearly highlight the potential benefit of including personality traits estimation inside the emotion recognition task.

One of the main contributions of this work constitutes the collection of the database. Hence, I would highly recommend that the authors provide more information and access to those data. In addition, they should complete the data availability statement, provided in the final sections of the MDPI manuscript templates.

I would suggest that the authors provided a more clear description of the paper’s contributions in the next to last paragraph of the introduction. Probably, a bulleted list could enhance the readability of this part.

References should be cited in the main body of the manuscript, using their corresponding numbers instead of the first author’s name. Please, again double-check the MDPI templates and correct citations.

The idea of modeling the long-term behavior of subjects in audio and image emotion recognition through CNN and/or LSTM architectures has been already investigated. Please refer to and discuss this motivation:

"Learning Long-Term Behavior through Continuous Emotion Estimation." The 14th PErvasive Technologies Related to Assistive Environments Conference. 2021.

"Attentive to Individual: A Multimodal Emotion Recognition Network with Personalized Attention Profile." Interspeech. 2019.

"Continuous Emotion Recognition for Long-Term Behavior Modeling through Recurrent Neural Networks." Technologies 10.3 (2022): 59.

"Personalised emotion recognition utilising speech signal and linguistic cues." 2019 11th International Conference on Communication Systems & Networks (COMSNETS). IEEE, 2019.

The data collection, acquisition and annotation as well as the methodology sections are both quite descriptive and easy to follow. My only comment is to add another subsection after the CNN and LSTM channels description that explains the fusion realized in the emotion classifier of Figure 2. The authors should clarify, what are exactly the inputs of this classifier (I assume the classification layers of CNN and LSTM are discarded before fusion), what are the dimensions of those before and after concatenation, as well as what is the output of the emotion classifier?

In line 333 a leave-one-subject-out cross-validation scheme is discussed. Yet, it is not clear if such a validation strategy is adopted only for the LSTM or the entire validation procedure of the system. Please, clarify the above and if the second case is valid, add a small paragraph regarding this validation scheme in the evaluation subsection. Also, refer to a work introducing this scheme:

Kansizoglou, Ioannis, Loukas Bampis, and Antonios Gasteratos. "An active learning paradigm for online audio-visual emotion recognition." IEEE Transactions on Affective Computing 13.2 (2019): 756-768.

Table 2 is a bit counter-intuitive. If it constitutes a confusion matrix, either the rows or columns of the table should sum up to 100%. Please double-check.

In the discussion section, the authors could discuss a bit more the fact that the exploitation of personality traits estimation led to enhance emotion recognition performance by comparing Tables 1 and 4.

There is no conclusions section. Please, provide a section concluding the main findings, benefits and limitations of your work as well as ideas for future work. If some of the above are already discussed in Section 4 please move them to the conclusion section.

Comments on the Quality of English Language

Please double-check for typos.

Minor comments

Lines 148-159: “a commonly used measure of personality that assesses.” Please rephrase.

Lines 197-198: “the order of stimuli was randomized for each person.” => “The order of stimuli was randomized for each person.”

Reviewer 4 Report

Comments and Suggestions for Authors

This paper study deep learning methods for emotion recognition using EEG signals.

The topic is very interesting and the results are sufficient.

I would suggest to discuss some latest deep neural network architecture, as the VGG model is relatively old.

The personality factor, such as the so called big five, should be further explained with more support from related references.

The computational efficiency and practical use of the proposed model should be discussed.

Comments on the Quality of English Language

The language is fine.

Round 2

Reviewer 1 Report

Comments and Suggestions for Authors

I thank the authors for addressing my previous suggestions and concerns. One issues, however, remains. The authors should check the ROC plots presented in Figure 6. The curves should be drawn using multiple points rather than a single point. Use class probabilities rather than class predictions for constructing ROC plot.

Reviewer 2 Report

Comments and Suggestions for Authors

The authors have revised the paper and, in my suggestion, it is essential to incorporate the aspect of future work into the revised version. This inclusion will ensure that the paper acknowledges and addresses potential areas of further research and development.

Comments on the Quality of English Language

There are still some grammatical errors, please check and make the corrections. 

Reviewer 3 Report

Comments and Suggestions for Authors

The authors have made great work addressing most of my comments. Therefore, I am happy to recommend the publication of the manuscript. Please, correct the following minor comments.

1) Pay careful attention to the figure and Table names and their references within the main body of the manuscript.
 - There are two Figures with the number 3, please correct them.
 - Other tables are cited in bold and others are not. Please, be concise with the MDPI templates.
 - In line 269, no Figure number is included and the space is too large.
 - On page 17, the space is too large. Please, modify accordingly.

2) I strongly believe, and after your further explanations in lines 482-484, that Table 5 is not a confusion matrix but four columns indicating the true positives (TP), true negatives (TN), false positives (FP), and false negatives (FN) for each class. The above is not the same as the confusion matrix. I strongly suggest that the authors remove the term confusion matrix and rename each column name with the corresponding TP, TN, FP and FN terms. Please, correct the above, because it is misleading to use the term confusion matrix which has a specific terminology and methodology.

In case the authors have more doubts, I advise them to search for the confusion matrix calculation methodology. It constitutes the calculation for each class (row) of its true positives as well as the false negatives (indicating where they have been classified among the remaining classes). Hence, each row sums to 1 (or 100% in the case of percentages).
